# Feeling Connected to Nature Attenuates the Association between Complicated Grief and Mental Health

**DOI:** 10.3390/ijerph21091138

**Published:** 2024-08-28

**Authors:** Madison Schony, Dominik Mischkowski

**Affiliations:** 1Department of Psychology, Ohio University, Athens, OH 45701, USA; 2Department of Psychology, University of Illinois, Champaign, IL 61820, USA; dominikm@illinois.edu

**Keywords:** grief, complicated grief, connection to nature, depression, anxiety

## Abstract

Complicated grief (CG) predicts decreased mental health over time. Furthermore, feeling connected to nature (CN) is positively associated with beneficial mental health outcomes, such as psychological wellbeing and perceived psychological resilience. Thus, we hypothesized that CN moderates the association between general grief or CG and negative mental health for bereaved people. Further, we hypothesized that one’s physical exposure to nature—that is, estimated time spent in nature and greenness (i.e., vegetation) surrounding one’s residential area—might moderate the association between general grief or CG and negative mental health for bereaved people. To test these hypotheses, we conducted a cross-sectional study. We sampled 153 participants who experienced the death of a close other by COVID-19 infection. Participants reported CG, general grief, anxiety symptoms, depression symptoms, CN, estimated time spent in nature, and residential area postal code via a single online survey. We estimated greenness surrounding participants’ residential areas using their self-reported five-digit U.S. postal code. Cross-sectional analyses indicated that, as predicted, CN attenuated the association between CG and depression, trended toward moderating the association between CG and anxiety, and did not moderate the associations between general grief and depression or anxiety. Other variables related to the experience of nature—the estimated time an individual spends in nature and the greenness surrounding one’s residential area—did not moderate the association between general grief or CG and depression or anxiety. We thus conclude that a sense of feeling connected to nature—not simply spending more time in nature or being surrounded by nature—may serve an important role in the mental health status of people experiencing complicated grief, perhaps because CN replenishes general belonging when someone significant has passed away.

## 1. Overview

In March 2021, the World Health Organization (WHO) declared COVID-19 a worldwide pandemic [1]. Merely a year later, the WHO reported 2.6 million deaths worldwide by COVID-19 infections [2]. By June 2024, the death toll of COVID-19 infections in the United States alone had reached 1.2 million [3]. COVID-19 deaths not only impacted those that died by the infection, but also the people who experienced the death of someone significant because of COVID-19 (i.e., experienced bereavement) [4]. Specifically, researchers have estimated that for every U.S. COVID-19 death, nine individuals are left bereaved [4]. Accordingly, millions of Americans must have experienced COVID-19-related bereavement.

General grief is a common, non-pathological reaction to bereavement and is typically marked by feelings of anger, guilt, and sadness; feelings of longing; social withdrawal; frequent crying; and frequent thoughts about the deceased [5]. General grief, itself, is not an illness, but instead an expected reaction to loss [6,7]. In contrast, grief that deviates from the normal process of accepting loss and adapting to life without the deceased is often categorized as pathological [8]. The International Classification of Diseases [9] and the *Diagnostic and Statistical Manual of Mental Disorders* [10] specify criteria for Prolonged Grief Disorder (PGD) as a form of pathological grief. PGD is colloquially termed ‘complicated grief’ [8]. Complicated grief involves a persistent grief response that causes significant impairment in personal leisure activities, social activities, occupational abilities, or the ability to form and maintain close relationships [9]. Characteristics of COVID-19 deaths (e.g., close others being unable to be physically present at the time of death) are likely to elicit a complicated grief response [11,12,13,14]. Thus, researchers have forecasted a widespread increase in complicated grief responses resulting from the COVID-19 pandemic [12,13]. Moreover, individuals that experience complicated grief report decreased mental health relative to individuals that do not experience complicated grief [15]. Decreased mental health, in turn, may lead to an elevated risk of physical illnesses, depression, anxiety, other psychiatric disorders, and mortality, particularly by suicide [16]. Together, complicated grief and associated negative mental and physical health outcomes likely pose a major public health concern.

Drawing upon literature demonstrating the mental health benefits of experiencing nature (e.g., [17,18]), feeling connected to nature (i.e., feeling affectively connected to nature based on prior experiences in natural settings [19]), or physical exposure to nature could mitigate the link between general grief or complicated grief and adverse mental health outcomes. The degree to which an individual feels connected to nature is positively associated with various indices of mental health, such as perceived psychological resilience [20]. Moreover, simply experiencing nature is associated with increased positive mood and decreased stress [17]. Finally, neighborhood greenness is associated with lower depression, anxiety, and stress symptoms [18]. Consequently, we sought to determine whether feeling connected to nature, time spent in nature, or residential greenness attenuate the association between general grief or complicated grief and mental health indicators related to depression and anxiety to gain a better understanding of grief and mental health, as well as of psychological factors that moderate the association between these two sets of variables.

## 2. Introduction

### 2.1. The COVID-19 Pandemic Likely Increased Rates of Complicated Grief

People are at risk for complicated grief when they cannot prepare for the death of a loved one (such as in the case of unexpected loss), cannot be physically present at the time of death, feel like they are unable to comfort the deceased, or disagree with or are uncertain about the best course of medical care during the dying process [21,22]. Common characteristics of many COVID-19 deaths map directly onto these risk factors for developing complicated grief and thus are likely to elicit a complicated grief response [12,13]. Specifically, people that lost loved ones because of COVID-19 experienced their loss as unexpected more often than people bereaved by other natural causes [12]. Furthermore, many loved ones of hospitalized patients that died because of COVID-19 were unable to be physically present at the time of death; a majority of hospitals prohibited visitors during the COVID-19 pandemic in order to prevent infection spread [23]. Finally, during the early stages of the COVID-19 pandemic, scientists lacked an understanding of COVID-19 disease course and treatment [24]. Consequently, treatment for COVID-infected patients was limited to palliative care or exploratory drug therapies before the U.S. Food and Drug Administration (FDA) approved drugs for COVID treatment in October 2020 [24,25]. This lack of knowledge about the typical course of the disease or effective treatment methods likely created uncertainty about appropriate medical care in individuals facing the loss of loved ones because of COVID-19. Such uncertainty about medical care has the potential to become a focus of rumination, derailing the grieving process and increasing the risk of developing complicated grief [6]. Moreover, COVID-19 social distancing and quarantining practices likely resulted in absent or non-traditional (e.g., virtual) death rituals, complicating the grieving process for many people [11,12]. Indeed, individuals bereaved because their loved one was infected with COVID-19 tend to experience more symptoms of PGD than individuals grieving deaths by other natural causes, which led researchers to predict widespread pandemic-related increases in complicated grief soon after the pandemic started [12].

### 2.2. General Grief and Complicated Grief Negatively Impact Mental Health

General grief and complicated grief responses to the passing of a loved one because of COVID-19 infection in turn may contribute to widespread, bereavement-related diminished mental health. Factor analyses reveal that general grief [26] and complicated grief [27] are distinct from depression and anxiety. Yet, both general grief and complicated grief are comorbid with depression and anxiety [28,29]. Specifically, those experiencing greater severity of grief burden reported greater depression severity [30]. Moreover, greater grief intensity is associated with greater levels of state anxiety [31].

However, complicated grief seems to have a stronger negative association with mental health than general grief: Individuals that experience complicated grief report lowered mental health after 6 months of bereavement compared to individuals experiencing general grief alone [15]. Given that complicated grief predicts decreased mental health over time and complicated grief may be on the rise, we expect that bereavement-related depression and anxiety may pose increased risk to public health. Indeed, anxiety and depression prevalence rose during the COVID-19 pandemic. Anxiety prevalence rates rose from 13% prior to the COVID-19 pandemic to 25–35% during the COVID-19 pandemic [32]. Similarly, depression symptom prevalence was higher at every level of severity (i.e., mild, moderate, moderately severe, severe) during the COVID-19 pandemic compared to before the COVID-19 pandemic (e.g., mild depression prevalence rose from 16.2% to 24.6% [33]). Though other factors are likely involved in these increases in anxiety and depression, bereavement may have played a key role in increased anxiety and depression rates during the COVID-19 pandemic.

### 2.3. Feeling Connected to Nature, Physical Time Spent in Nature, and Being Surrounded by Nature Are Positively Associated with Mental Health

Despite ample evidence for the general mental health benefits of experiencing nature, it remains unknown which aspect of experiencing nature may be particularly important for the mental health status of grieving persons. It is possible that an individual’s simple sense of belonging to or feeling connected to nature independently of their actual physical exposure to nature may be a moderating factor. Meta-analytic results demonstrate that feeling connected to nature is positively associated with overall well-being, psychological well-being (e.g., purpose/meaning in life), emotional well-being (e.g., positive affect, life satisfaction), and social well-being (e.g., social acceptance, social integration), with large effect sizes shown for overall and psychological well-being [34]. Further, feeling connected to nature at least partially mediates increases in positive affect following nature exposure [35] and may decrease negative rumination [17]. Thus, feeling connected to nature may mitigate the link between general grief or complicated grief and reduced mental health by increasing overall well-being and reducing rumination about personal loss.

In addition to simply feeling connected to nature, variables related to physical exposure to nature are also associated with better mental health. Specifically, self-reported time spent visiting nature (e.g., hours per day) and self-reported frequency of nature visits (e.g., visits per month) are positively associated with greater overall mental health [36,37]. Researchers have also used the greenness (i.e., vegetation) surrounding people’s living spaces as an objective measure of physical exposure to nature. Vegetation indices combine reflectance levels of visible light (which is absorbed by vegetation) and near-infrared light (which is reflected by vegetation) into ratios to quantify the vegetation of a particular area [38]. Research using vegetation indices demonstrates that living in a greener environment is associated with mental health benefits [18,39]. Thus, physical exposure to nature may also affect the link between general grief or complicated grief and mental health status.

A first step to buffer bereaved individuals from experiencing reduced mental health associated with complicated grief is to identify factors that mitigate the link between complicated grief and adverse mental health consequences. Both feeling connected to nature and actual exposure to nature—operationalized either as time spent in contact with nature or habitual closeness to green environments—are positively correlated with mental well-being [17,40,41]. However, it has not yet been established whether and how varying degrees of feeling connected to nature, time spent in nature, or residential greenness relate to the mental health status of those experiencing complicated grief. Possibly, feeling more connected to nature or spending more time in or near nature may be associated less with decreased mental health comorbid with many psychiatric conditions (e.g., complicated grief). To gain a better understanding of the association between grief and mental health, we sought to examine whether nature-related variables (i.e., feeling connected to nature, time spent in nature, residential surrounding greenness) attenuate the association between general grief or complicated grief and indicators of mental health, specifically, depression and anxiety.

## 3. Methods

### 3.1. Procedure

We recruited bereaved participants online by posting advertisements on mTurk, Research Match, and grief group pages on Facebook (i.e., “COVID-19 Loss Support for Family & Friends”) and Reddit (i.e., “r/COVIDgrief”). We sampled from diverse online data collection platforms to minimize sampling bias. We recruited participants between 10 March 2021 and 30 January 2022. Participants recruited from mTurk, Facebook, and Reddit accessed the survey through a link posted on the respective sites. Research Match participants received an initial contact email without a survey link. Once Research Match participants indicated the desire to be contacted further, we sent a second email containing the survey link.

After providing informed consent, participants confirmed that they had lost a close other to COVID-19 infection and were willing to provide verification of the passing. Participants then identified the name and passing date of the close other who had passed. Afterward, we prompted participants to either provide an online link (e.g., to an online obituary, to an online death announcement) or upload other verifying documentation (e.g., an obituary, a screenshot of death announcement). Finally, participants completed the survey measures.

We deemed responses valid if the name and date of passing reported by participants matched the information provided in the online link or uploaded documentation. Participants were excluded from the study if they reported that they had not experienced the passing of a close other, were not willing to provide verification of the passing, did not provide verification documents matching the name and date of the person who had passed, were not 18 years or older, or lived outside the United States (see Figure 1). We intended to identify fraudulent online participants (i.e., participants concealing a device location outside the United States using virtual private servers to conceal internet provider addresses [42]) by asking participants to name a pictured vegetable (i.e., an eggplant). An eggplant is frequently identified by other names (e.g., brinjal) in areas outside of the United States, such as in India, where estimates of 40% of mTurk workers reside [43]. Therefore, we excluded data from analyses when participants indicated a term other than eggplant or did not indicate a term as a cautionary measure to remove fraudulent data from the study (see Figure 1). 

Participants who completed the survey received a USD 1 (mTurk) or USD 2 (Research Match, Reddit, Facebook) Amazon gift certificate. Compensation differed between recruitment platforms in accordance with platform compensation standards [44,45]. We compensated participants from recruitment platforms without an established compensation standard (i.e., Research Match, Reddit, Facebook) based on Ohio minimum wage (USD 7.25/h). The Ohio University institutional review board approved all study procedures.

### 3.2. Measures

#### 3.2.1. Grief

Complicated Grief. Participants completed the Brief Grief Questionnaire (BGQ) [46], which is a short screener for complicated grief. The screener was developed for measuring complicated grief symptoms among crisis counseling recipients following the 11 September 2001 terrorist attacks [46] and has since demonstrated sufficient reliability and discriminant validity [47]. The BGQ consists of five items assessing respondent experiences with the following symptoms: intrusive images of the deceased, avoidance of reminders of the deceased, trouble accepting the death, feelings of detachment from others, and overall interference with life. Participants indicated the extent to which each item applied to them using a 3-point scale from 0 (not at all) to 2 (a lot). Cut-off values for determining complicated grief presence are not well established [46,47]. Therefore, we used responses to the BGQ as a continuous variable. We summed items to create a measure of complicated grief (α = 0.81), with higher scores indicating greater complicated grief.

General Grief. Participants completed a measure of general grief using the Bereavement Experience Questionnaire (BEQ-24) [48]. Participants responded to 24 items, such as, “Since the death of this person, I have yearned for the deceased person”, on scales from 1 (not at all) to 4 (very much). We did not group participants into groups (e.g., high grief symptomatology, low grief symptomatology) based on the BEQ-24. Instead, we averaged items into a measure of general grief (α = 0.94), with greater scores indicating greater general grief.

#### 3.2.2. Mental Health

Anxiety. We assessed participants’ anxiety using the Generalized Anxiety Disorder Screener (GAD-7) [49]. This screener has demonstrated sufficient reliability and validity as a measure of anxiety in the general population [50]. Participants indicated the frequency with which they had experienced seven anxiety symptoms, such as “Not being able to stop or control worrying”, in the past 2 weeks on scales ranging from 0 (not at all) to 3 (nearly every day). We summed items to create a measure of anxiety (α = 0.95) with higher scores indicating greater anxiety.

Depression. Participants indicated the extent to which they experienced depressive symptoms in the past week using the PROMIS Short Form v1.0 Depression 8a [51]. This short-form screener has proven reliable and valid in diverse populations [52]. Participants rated eight depression-related symptoms, such as “I feel hopeless”, on scales ranging from 1 (never) to 5 (always). We summed items into a raw score which we then transformed into a standardized t-score (*M* = 50, *SD* = 10) for each participant using the appropriate score conversion chart [53]. Participant’s standardized t-score served as a measure of depression (α = 0.96) with higher scores indicating greater depression.

#### 3.2.3. Nature-Related Variables

Feeling Connected to Nature. Participants indicated their affective connection to nature based on prior experiences in natural settings using Mayer and Frantz’s (2004) Connectedness to Nature scale (CNS) [19]. Participants indicated their agreement with 13 items (one item of the original CNS was accidentally omitted; internal consistency among items remained acceptable as measured by Cronbach’s alpha, suggesting that the included 13 items were sufficient for measuring the targeted construct), such as, “I often feel a kinship with animals and plants,” on a scale from 1 (strongly disagree) to 5 (strongly agree). We averaged items to create a measure of feeling connected to nature (α = 0.80), with higher scores indicating feeling more connected to nature.

Time Spent in Nature. Participants estimated the amount of time they typically spend in nature using items modified from Beyer and colleagues [54] and Larson and colleagues [55]. Participants estimated the typical amount of time spent outdoors in nature each day of the week, each day of the month, and each day when the weather permits on scales from 1 (30 min or less) to 7 (more than 5 h). We summed items to create a time spent in nature measure (α = 0.90), with higher values indicating more time spent in nature.

Surrounding Greenness. We estimated greenness surrounding participants’ residential areas using the 5-digit U.S. postal code reported by participants. First, we identified the geographical spaces associated with postal code areas, which were irregularly shaped. Then, we fitted rectangles around these geographical spaces. Next, we used the latitudinal and longitudinal coordinates of the centermost point as well as the length and width of these rectangles to define the geographical areas nearest to where participants lived. Finally, we calculated Normalized Difference Vegetation Index (NDVI) values of these rectangles to estimate the vegetation level of participants’ residential areas.

The NDVI is a vegetation index obtained via satellite images from land surface reflections of near-infrared and visible wavelengths [56]. Chlorophyll, a green pigment crucial for photosynthesis in plants and thus prominent in densely vegetative areas, absorbs visible light but reflects near-infrared light [57]. NDVI values represent the ratio of the difference between near-infrared and visible light to the sum of near-infrared and visible light reflected from a given surface area (i.e., $\frac{NearInfraredLight-VisibleLight}{NearInfraredLight+VisibleLight}$ [57]). In areas with more vegetation, more chlorophyll is present and, thus, more visible light is absorbed, causing the numerator of this ratio to be larger and, thus, the resulting NDVI value to be closer to 1. Therefore, NDVI values will be higher in more vegetative areas [57]. In general, NDVI values range from −1 to 1, with values closer to −1 indicating water bodies, 0 indicating no vegetation, and 1 indicating high vegetation [56,57].

We extracted NDVI values for each postal code area using the MODIS Global Subsets Tool (https://modis.ornl.gov/globalsubset, accessed on 22 August 2022), similar to previous research on postal code vegetation [58]. This tool utilizes data from MODerate-resolution Imaging Spectroradiometer (MODIS) satellite images at 250 m resolution taken every 16 days. We averaged mean NDVI values in the postal code area for each image taken between the date the participant took the survey and a year prior to the participation date. If an image was not taken exactly on the date a year prior to a survey response, we used the mean NDVI value from the image that was taken at the next date. Higher NDVI values thus indicate greater annual average vegetation (i.e., surrounding greenness) in rectangular areas surrounding participants’ U.S. postal code regions.

### 3.3. Data Analysis

We analyzed data in SPSS 28 [59]. First, we calculated Pearson’s correlations between study variables. Next, we conducted simple moderation analyses to test study hypotheses. Namely, we tested whether feeling connected to nature moderated the associations between complicated grief and depression, between complicated grief and anxiety, between general grief and depression, and between general grief and anxiety. Further, we tested whether time spent in nature moderated the associations between complicated grief and depression, between complicated grief and anxiety, between general grief and depression, and between general grief and anxiety. Finally, we tested whether surrounding greenness moderated the associations between complicated grief and depression, between complicated grief and anxiety, between general grief and depression, and between general grief and anxiety. We further probed any significant (*p* < 0.05) or marginally significant (*p* < 0.10) simple moderation by conducting simple slope analyses, examining the effect of complicated grief or general grief on depression or anxiety at varying levels (i.e., 1 standard deviation below the mean, the mean, and 1 standard deviation above the mean) of nature-related variables.

## 4. Results

Participants included 153 bereaved individuals (109 females; *M*_age_ = 42.1, *SD* = 13.8; 114 White/European Americans, 15 Hispanics/Latinos, 11 Black/African Americans, 4 Asian Americans/Pacific Islanders, and 9 of unidentified race/ethnicity) who had lost a close person due to COVID-19. We recruited 87 participants from mTurk, 45 from Research Match, 15 from grief groups on Facebook, and 6 from grief groups on Reddit. We recruited as many participants as possible from various platforms over a 10-month period while the survey was available. The sample size, thus, reflects the number of individuals we were able to access who had experienced COVID-19 bereavement and provided sufficient data for us to deem their responses as valid.

Not all participants completed all measures in full. In some cases, participants did not provide any response on measures of anxiety (*n* = 1), depression (*n* = 1), time spent in nature (*n* = 2), or U.S. postal code (*n* = 3). We were not able to locate one participant’s postal code, therefore constituting an additional missing value for surrounding greenness. In some cases, participants answered all but one question on measures of complicated grief (*n* = 1), general grief (*n* = 4), anxiety (*n* = 2), depression (*n* = 2), feeling connected to nature (*n* = 3), and time spent in nature (*n* = 1). Further, two participants answered all but two questions on the measure of general grief. Nevertheless, we retained these participants when they had provided sufficient data (i.e., answered most questions for a measure) for a particular set of analyses. We did not substitute or impute data in cases of missing data.

Though we recognize the current sample may not be perfectly comparable to other samples, descriptive statistics of reported general grief (*M* = 1.91, *SD* = 0.69) indicated that our participants experienced average levels of general grief comparable to samples of young adults bereaved by parental death (e.g., [60]). In contrast, our participants reported elevated levels of complicated grief (*M* = 4.45, *SD* = 2.74) compared to samples of cancer-bereaved and Japanese nonspecific-bereaved people [47,61]. Finally, participants reported elevated levels of anxiety (*M* = 7.40, *SD* = 6.41) [50,53] and depression (*M* = 55.51, *SD* = 10.78) [62]. Though the mean anxiety level in the sample reflected above-average levels of anxiety, mean anxiety scores in the sample reflected mild levels of anxiety remaining within non-clinical ranges [49,52,63]. Similarly, mean depression levels in the sample reflected mild to moderate levels of depression, nearing a value (i.e., 60) indicating clinical significance [62]. Table 1 shows the Pearson correlations between the primary study variables.

To test whether feeling connected to nature moderated the association between complicated grief and depression, we conducted a simple moderation analysis in SPSS 28 [59] using the PROCESS macro model 1 [64]. Indeed, feeling connected to nature attenuated the association between complicated grief and depression, *b* = −1.36, *SE =* 0.58, *t*(148) = −2.34, *p* = 0.021, 95% CI [−2.50, −0.21] (see Figure 2). We probed this moderation by conducting simple slope analyses of the association between complicated grief and depression at different levels of feeling connected to nature. There was a significant association between complicated grief and depression at low levels of feeling connected to nature (*b* = 2.23, *SE =* 0.44, *t*(148) = 5.01, *p* < 0.001, 95% CI [1.35, 3.11]) and medium levels of feeling connected to nature (*b* = 1.49, *SE =* 0.30, *t*(148) = 5.05, *p* < 0.001, 95% CI [0.91, 2.07]). In contrast, at high levels of feeling connected to nature, the association between complicated grief and depression was non-significant (*b* = 0.75, *SE =* 0.42, *t*(148) = 1.78, *p* = 0.08, 95% CI [−0.08, 1.58]).

Next, we tested whether feeling connected to nature also moderated the association between complicated grief and anxiety. We found tentative evidence for this idea. Specifically, feeling connected to nature attenuated the association between complicated grief and anxiety, though this moderation only approached conventional levels of significance, *b* = −0.61, *SE =* 0.34, *t*(148) = −1.78, *p* = 0.077, 95% CI [−1.29, 0.07] (see Figure 3). Similar to the pattern observed when testing the association between complicated grief and depression, simple slope analyses revealed a significant association between complicated grief and anxiety at low levels of feeling connected to nature (*b* = 1.30, *SE* = 0.26, *t*(148) = 4.94, *p* < 0.001, 95% CI [0.78, 1.82]) and medium levels of feeling connected to nature (*b* = 0.96, *SE* = 0.17, *t*(148) = 5.54, *p* < 0.001, 95% CI [0.62, 1.31]). At high levels of feeling connected to nature, the association between complicated grief and anxiety was weaker in size but remained significant (*b* = 0.63, *SE* = 0.25, *t*(148) = 2.54, *p* = 0.012, 95% CI [0.14, 1.12]).

In contrast, feeling connected to nature did not moderate the associations of general grief with depression or anxiety, *b*s ≤ −0.85, *SE*s ≥ 1.35, *t*s(148) ≤ −0.47, *p*s ≥ 0.47, though we found that general grief predicted depression and anxiety, *b*s ≤ 10.30, *SE*s ≤ 0.97, *t*s(150) ≤ 10.64, *p*s ˂ 0.001. Neither time spent in nature nor surrounding greenness moderated the associations of general or complicated grief with depression or anxiety, *b*s ≤ 7.06, *SE*s ≥ 0.04, *t*s ≤ 1.22, *p*s ≥ 0.22.

## 5. Discussion

Given previous evidence that feeling connected to nature [17,34], more time spent in nature [36], and more residential greenness (e.g., [65]) are associated with better mental health, we tested whether these variables moderated the association between grief and poor mental health following experiencing the loss of a close other by COVID-19 infection. Specifically, we provide evidence that feeling connected to nature attenuates the association between complicated grief and depression for people bereaved by the death of a close other because of COVID-19 infection. Further, we provide tentative evidence that feeling connected to nature also attenuates the association between complicated grief and anxiety, though more evidence is necessary to establish this finding. In contrast, we did not find evidence that mere physical exposure to nature (i.e., time spent in nature or residential greenness) affected the association between grief and mental health. We thus conclude that feeling connected to nature—not simply spending more time in nature or being surrounded by nature—may serve an important role in the mental health status of people experiencing complicated grief.

We suggest a reason for why feeling connected to nature rather than physical exposure to nature attenuated the association between complicated grief and depression and trended toward attenuating the association between complicated grief and anxiety: it is possible that feelings of belongingness to nature may serve as a proxy for reduced general feelings of belonginess and interconnectedness caused by losing loved ones. Relatedness—feeling connected and a sense of belongingness with others—is a basic psychological need according to self-determination theory [66,67]. Furthermore, the belongingness hypothesis suggests that humans are fundamentally motivated to satisfy their need to belong (NTB) by forming and maintaining lasting, positive, and significant interpersonal relationships [68]. One may satisfy such needs by engaging in frequent, affectively pleasant interactions with others, specifically those that show a stable and enduring concern for one’s welfare [68]. Experiencing an unmet NTB is associated with decreased well-being and increased anxiety and depression [69]. Therefore, in the case of bereavement, individuals will likely lose close others that served as important sources for satisfying their NTB, and they will also likely experience reduced mental health as a consequence. Perhaps, feeling connected to nature may disrupt the relationship between complicated grief and poor mental health outcomes by satisfying one’s NTB. Indeed, researchers have theorized that an individual’s NTB may be satisfied through a sense of belonging to or connectedness with nature [35]. In contrast, more time spent in nature or closer proximity to greener spaces may not necessarily be associated with greater feelings of belongingness, as physical exposure to nature may not automatically lead to a sense of connection to nature. More research testing this interpretation of our results is necessary to confirm a role of replenished general belongingness via feeling connected to nature in attenuating the association between complicated grief and mental health.

### 5.1. Theoretical and Practical Implications

Previous research has primarily focused on psychological (e.g., self-criticism) [70] and interpersonal factors (e.g., pre-loss dependency) [71] in understanding and treating complicated grief. Namely, numerous psychological and interpersonal risk factors for CG have been identified (e.g., unpreparedness for loss, family conflict at end of life) [20,21,72]. Research demonstrates that targeted therapeutic interventions for complicated grief—called ‘complicated grief treatment’ [73]—are effective at improving CG symptoms. Further, some research has focused on identifying protective factors against CG, with findings demonstrating that early preparation for the death of close others leads to lower levels of CG after loss [74]. However, researchers have focused less on mitigating negative mental health related to CG. That is, researchers have developed therapeutic interventions for CG and identified factors that may help to prevent CG; yet, there has been little focus on alleviating comorbid anxiety and depression once CG has occurred, with a few notable exceptions. Specifically, Glickman and colleagues [75,76] have identified factors that affect anxiety and depression comorbid with complicated grief: complicated grief treatment aids in reducing CG-related anxiety and depression [75], and the reduction in guilt, self-blame, negative thoughts of the future, and avoidant behaviors serve as mediators between complicated grief treatment and CG outcomes, namely complicated grief symptomatology and daily functioning [76]. The present research offers a novel perspective in identifying factors involved in the association between CG and reduced mental health by exploring nature-related factors that may moderate the association between CG and depression or anxiety. Such an approach is valuable given the difficulty of treating CG with traditional grief-focused therapeutic approaches [29], physician uncertainty about diagnosing CG or identifying appropriate treatments for CG [73], and barriers to therapeutic or pharmacological treatment for CG (e.g., when grieving people reside in highly rural areas). Given widespread accessibility to natural spaces—even for those residing in cities (e.g., parks), but especially for those residing in rural areas—we sought to demonstrate that nature-related factors were related to different associations between CG and mental health. Our findings suggest that nature-related factors may, in fact, be implicated in differences in the association between CG and mental health, extending work on identifying factors that affect anxiety and depression comorbid with complicated grief (e.g., [75]). Pending experimental work, practical implications may include mitigation of CG-related depression or anxiety by increasing one’s connection to nature via intervention. Accordingly, researchers should test whether incorporating nature-based interventions into grief therapy programs could enhance treatment outcomes by leveraging the therapeutic benefits of feeling connected to nature. Such research may draw upon literature demonstrating that therapists can nurture individuals’ connection to nature, such as through encouraging mindful engagement with nature, which may include mindfully listening to nature sounds, mindfully looking at nature near and far [77], and noting the good things in nature each day [78]. Overall, our suggestion of an important role of CN in moderating the relationship between CG and mental health generates further research questions that, once answered, may aid in mitigating the link between CG and decreased mental health.

### 5.2. Limitations and Future Directions

Our study also has limitations. First, causal interpretation of our data is limited due to the data’s correlational nature. Though we found that feeling connected to nature was negatively correlated with the association between complicated grief and depression, it remains unclear whether experimentally increasing connection to nature would attenuate the association between complicated grief and depression. Future research using an experimental approach to manipulate connection to nature—such as by participating in environmental education programs (e.g., [79], but see [80,81]) to attenuate the association between complicated grief and mental health would help to clarify whether feeling connected to nature is causally involved in moderating the association between complicated grief and mental health. In addition to experimental studies, longitudinal research may also be useful in understanding whether and how changes in feeling connected to nature alter the association between complicated grief and mental health over time.

Our reliance on correlational data also limits our understanding of the mechanisms by which feeling connected to nature attenuates the association between complicated grief and reduced mental health. Though we speculate that a stronger sense of feeling connected to nature may simultaneously promote stronger feelings of belongingness to affect mental health outcomes related to complicated grief, alternative or additional psychological mechanisms may be at play (e.g., CN may reduce emotional pain caused by CG, thereby affecting the relationship between CG and mental health). Additional research is necessary for testing such pathways (i.e., whether a sense of belongingness or reduced emotional pain via feeling connected to nature drives CN moderating the association between CG and depression) and thus clarifying the mechanism by which feeling connected to nature attenuates the association between complicated grief and mental health.

Another limitation of our study is having a non-representative sample. Specifically, our sample was largely made up of females and White- or European American-identifying individuals, limiting the generalizability of study conclusions to individuals outside of these demographics. Future research may explore the relationships among study variables with a sample more representative of the U.S. (e.g., including more males, more people of Black or African American racial identity) to expand the generalizability of findings.

Moreover, though we required participants submit verifying documents to corroborate self-reported loss, this study is limited by not requiring participants to corroborate self-reports of a close other passing with a death certificate. Despite not requiring death certificates, some participants uploaded death certificates, and the verification documents individuals were required to upload often corroborated self-reports of death by COVID-19 infection. In future research, the use of a death certificate to verify passing could help to corroborate self-reports of passing, cause of death, and close relation to the individual that passed.

Another limitation of the current study is that important study variables were based on participants’ self-report rather than direct observations of participants’ thoughts and behaviors. To obtain a sample of COVID-19-bereaved individuals, we opted for online data collection, given the large number of people accessible online. Due to collecting data online, we were not able to observe behaviors directly. For example, we relied on self-reported time spent in nature rather than direct observations of the time an individual spent in nature. However, we employed established measures (e.g., GAD-7) in an attempt to obtain valid data via self-report. Future research may address this limitation by employing different methods of data collection (e.g., ecological momentary assessment for time spent in nature, clinician evaluation of complicated grief presence), perhaps with bereaved individuals locally available to researchers.

Another potential limitation of the current study is the resolution at which surrounding greenness images were taken. The resolution at which surrounding greenness images were taken may have limited the precision by which we were able to estimate participant surrounding greenness. Namely, a higher resolution than the 250 m resolution of surrounding greenness images used in the current study could identify more subtle variations in vegetation. Therefore, researchers may consider using or constructing methods for surrounding greenness estimation with improved resolution.

Finally, our study may be limited in power, despite attempts to recruit an adequate sample of grieving persons. It is possible that a larger sample of bereaved individuals may provide greater power for detecting whether time spent in nature or surrounding greenness moderate the association between complicated grief and mental health. Additional research evaluating how nature-related variables of feeling connected to nature, time spent in nature, and surrounding greeness correlate with the association between complicated grief and mental health should thus collect larger samples or use more powerful research designs (e.g., repeated measures of CN, depression, and anxiety) for improving our understanding of the role of nature-related variables in altering the association between grief and mental health.

## 6. Conclusions

Our study contributes to the understanding of the associations between complicated grief and mental health among individuals bereaved by COVID-19 by identifying a novel factor that may attentuate this association: connection to nature. Building upon existing literature that highlights the detrimental effects of CG on mental health [15] and the beneficial impacts of CN on psychological wellbeing [17,18,36,39], our findings underscore the role of CN as a moderator in the association between CG and depression and anxiety. In contrast, we found that simply spending more time in nature or residing in greener environments did not moderate the association between CG and depression or anxiety. Instead, it is the subjective feeling of connection to nature—not time spent in nature or surrounding greenness—that appears to moderate this association suggesting that perhaps feeling connected to nature unlike physical exposure to nature may disrupt the relationship between complicated grief and poor mental health outcomes by satisfying one’s need to belong. Thus, our study highlights the potential importance of CN in shaping the mental health of people experiencing complicated grief.

## Figures and Tables

**Figure 1 ijerph-21-01138-f001:**
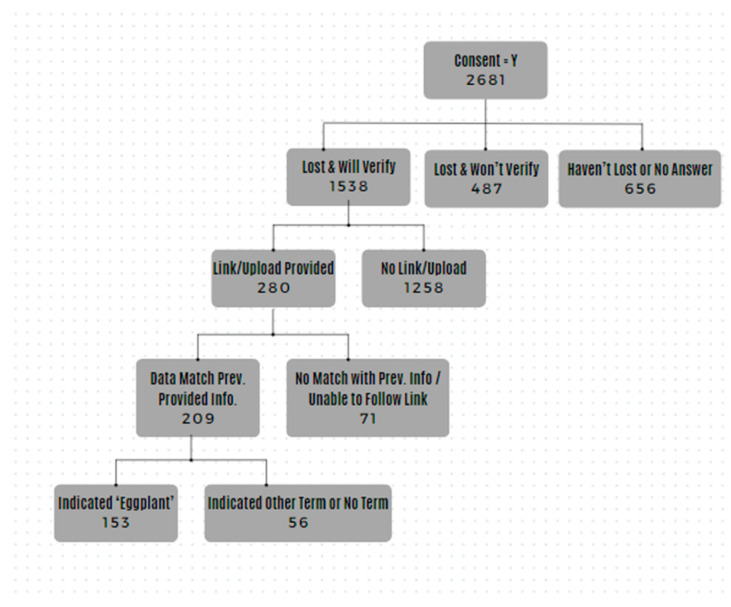
Flow diagram of participant exclusion.

**Figure 2 ijerph-21-01138-f002:**
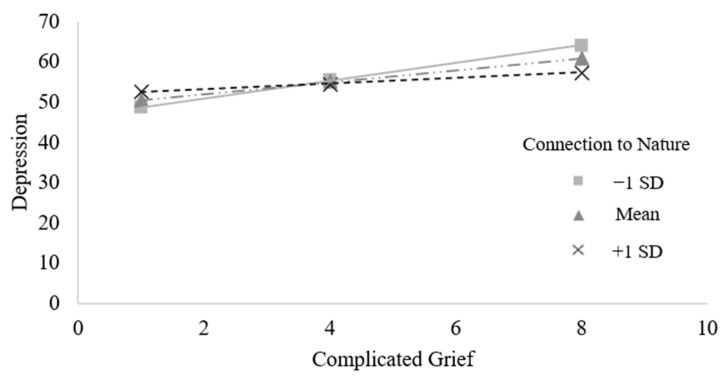
Connection to nature attenuated the association between complicated grief and depression. Slopes depict the association between complicated grief and depression at 1 standard deviation below the mean (i.e., −1 SD), the mean, and 1 standard deviation above the mean (i.e., +1 SD) of connection to nature.

**Figure 3 ijerph-21-01138-f003:**
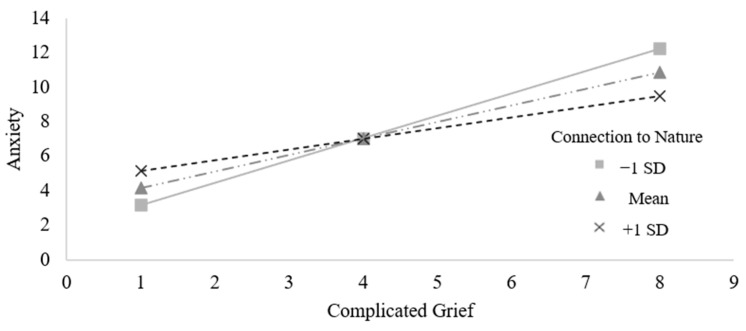
Connection to nature marginally attenuated the association between complicated grief and anxiety. Slopes depict the association between complicated grief and anxiety at 1 standard deviation below the mean (i.e., −1 SD), the mean, and 1 standard deviation above the mean (i.e., +1 SD) of connection to nature.

**Table 1 ijerph-21-01138-t001:** Pearson correlations between the primary study variables.

Variable	1	2	3	4	5	6
1. Complicated Grief						
2. General Grief	0.70 *					
3. Anxiety	0.40 *	0.64 *				
4. Depression	0.37 *	0.66 *	0.72 *			
5. Connectedness to Nature	0.06	−0.03	−0.01	−0.04		
6. Time Spent in Nature	0.30 *	0.22 *	0.15	0.11	0.26 *	
7. Surrounding Greenspace	0.12	−0.05	−0.04	−0.11	0.04	0.06

* *p* < 0.01.

## Data Availability

Data is contained within Appendix A.

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
