# Peer review of "Feeling Connected to Nature Attenuates the Association between Complicated Grief and Mental Health"

_ijerph, 2024, doi:10.3390/ijerph21091138_

Round 1

Reviewer 1 Report

Comments and Suggestions for Authors

This research paper addresses the feeling Connected to Nature Attenuates the Association between Complicated Grief and Mental Health, below are my comments to the authors

·        I could not find any paragraph about the data analysis!

·        How the sample size was arrived at? Any sample size calculations?

·        The methods-procedures- mentions the number of participants included, this should be transferred to the results with a flow chart figure explaining it

·        The data collection refers to 2022, how can be the data collected be correlated to deaths due to COVID? Couldn’t the grief be due to other factors?, why there was no control group of non-bereaved people.

·        The tables should mention the statistical analysis performed

·        The data was collected online, how did the authors ensure to minimize the bias in the data?

Author Response

Thank you for taking the time to review this manuscript. Please find the detailed responses below and the corresponding revisions highlighted in the re-submitted files.

Comment 1: “I could not find any paragraph about the data analysis!”

Response 1: Instead of readers having to decipher data analysis procedures from the results section alone, we agree that the addition of a data analysis paragraph would make reading the results section clearer. Therefore, we have added a data analysis paragraph. This change can be found within section 2.3 of the uploaded revised manuscript.

Comment 2: “How the sample size was arrived at? Any sample size calculations?”

Response 2: We have added a statement about sample size: “We recruited as many participants as possible from various platforms over a 10-month period while the survey was available. The sample size, thus, reflects the number of individuals we were able to access who had experienced COVID-19 bereavement and provided sufficient data for us to deem their responses as valid.” We have also added Figure 1, which is a flow diagram of the number of participants at each stage of the study and reasons for non-eligibility at each stage of the study. This figure shows that a total of 2,681 participants consented to the study, but only 153 responses were deemed eligible, helping to explain our efforts of recruiting a large sample with the time and monetary resources available to us, but demonstrating the lack of accuracy in bereavement-related data collection online which limited our resulting sample size. 

Comment 3: “The methods-procedures mentions the number of participants included, this should be transferred to the results with a flow chart figure explaining it.”

Response 3: We moved the information about the number of participants included to the results section. Further, we included a flow chart explaining the number of participants at each stage of the study and reasons for non-eligibility at each stage of the study (see Figure 1).

Comment 4: “The data collection refers to 2022, how can be the data collected be correlated to deaths due to COVID? Couldn’t the grief be due to other factors?, why there was no control group of non-bereaved people.”

Response 4: The data collection period was from March 10, 2021 to January 30, 2022. During this time, after consenting to partake in the study, participants had to specifically answer a question about whether they had experienced the passing of a close other due to COVID-19 illness (see 2.3. Data Analysis). We have added the following statement to 4.2. Limitations & Future Directions: “Moreover, though we required participants submit verifying documents to corroborate self-reported loss, this study is limited by not requiring participants to corroborate self-reports of close-other passing with a death certificate. Despite not requiring death certificates, some participants uploaded death certificates and the verification documents individuals were required to upload often corroborated self-reports of death by COVID-19 infection. In future research, the use of a death certificate to verify passing could help to corroborate self-reports of passing, cause of death, and close relation to the individual that passed.” There was no control group of non-bereaved people as the study design allowed for continuous levels of general grief and complicated grief, no experimental manipulations were conducted, and analyses were cross-sectional, comparing at an individual time point how nature-related variables affect the relationship between continuous variables of general grief or complicated grief and continuous variables of anxiety and depression. No grouping was completed, allowing for data analysis without a non-bereaved control group.

Comment 5: “The tables should mention the statistical analysis performed.”

Response 5: Thank you for catching this oversight. We have corrected our Table 1 description to specify that Pearson correlations between primary study variables were performed, rather than just generally indicating correlations among study variables were performed.

Comment 6: “The data was collected online, how did the authors ensure to minimize bias in the data?”

Response 6: One method we used to minimize bias in the data was to sample participants from different online data collection platforms. We now state this in the manuscript: “We sampled from diverse online data collection platforms to minimize sampling bias.” Sampling from different online data collection platforms helps to minimize sampling bias by providing many, diverse people accessibility to study participation and limiting systematic inclusion of members of the population into the study. Another method we used to minimize bias in the data was to identify potentially fraudulent online participants. We detail this in the manuscript now by stating: “We intended to identify fraudulent online participants (i.e., participants concealing a device location outside the United States using virtual private servers to conceal internet provider addresses) by asking participants to name a pictured vegetable (i.e., an eggplant). An eggplant is frequently identified by other names (e.g., brinjal) in areas outside of the United States, such as in India where estimates of 40% of mTurk workers reside. Therefore, we excluded data from analyses when participants indicated a term other than eggplant or did not indicate a term as a cautionary measure to ensure data from fraudulent online participants were not included in the study (see Figure 1).” Finally, we used continuous mental health variables (rather than categorizing by presence of condition) and not substituting or imputing data in cases of missing data to minimize bias. The manuscript reads, for each of these methods respectively: “Therefore, we used responses to the BGQ as a continuous variable. We summed items to create a measure of complicated grief (α = .81) with higher scores indicating greater complicated grief” and “We did not substitute or impute data in cases of missing data.”

Reviewer 2 Report

Comments and Suggestions for Authors

I consider that it is a very interesting investigation that opens a perspective to improve mental health with contact with nature. The following changes should be made: > the abstract should clarify what type of study it is

use strobe as a checklist and attach it to the study

I consider that the sample is not representative of ethnic minorities, explain this in the limitations of the study, another limit that should be addressed is that the death certificate is absent to corroborate the data given, self-reports with the connection with nature is another limit that should be indicated, we do not know how much specific time outdoors, etc., the NDVI is very interesting but the resolution at 250m may not have identified more subtle variations in vegetation, describe in more detail the missing data of the participants, other questions to be resolved are how the possible variability of interpreting the questions was managed

Author Response

Thank you for taking the time to review this manuscript. Please find the detailed responses below and the corresponding revisions highlighted in the re-submitted files.

Comment 1: “The abstract should clarify what type of study it is.”

Response 1: We have clarified that the study is a cross-sectional study in the abstract. A portion of the abstract now reads: “To test these hypotheses, we conducted a cross-sectional study.”

Comment 2: “Use STROBE as a checklist and attach it to the study.”

Response 2: We have now used a STROBE checklist to clarify all elements are included in the manuscript. The STROBE checklist is now attached to the study as a Supplementary File.

Comment 3: “I consider that the sample size is not representative of ethnic minorities, explain this in the limitations of the study.”

Response 3: We agree that the sample size is not representative of ethnic minorities within the United States. We have added this as a limitation of the study in 4.2. Limitations & Future Directions. We now state the following: “Another limitation of our study is having a non-representative sample. Specifically, our sample was largely made up of females and White or European American identifying individuals, limiting the generalizability of study conclusions to individuals outside of these demographics.”

Comment 4: “Another limit that should be addressed is that the death certificate is absent to corroborate the data given.”

Response 4: We agree that the lack of death certificates to corroborate self-reports is a limitation of this study. We have added this limitation of the study in 4.2. Limitations & Future Directions. We state: “Moreover, though we required participants submit verifying documents to corroborate self-reported loss, this study is limited by not requiring participants to corroborate self-reports of close-other passing with a death certificate. Despite not requiring death certificates, some participants uploaded death certificates and the verification documents individuals were required to upload often corroborated self-reports of death by COVID-19 infection. In future research, the use of a death certificate to verify passing could help to corroborate self-reports of passing, cause of death, and close relation to the individual that passed.”

Comment 5: “Self-reports with the connection with nature is another limit that should be indicated. We do not know how much specific time outdoors, etc.”

Response 5: We have added this limitation of the study in 4.2. Limitations & Future Directions. We state: “Another limitation of the current study is that important study variables are based on participants’ self-report rather than direct observations of participants’ thoughts and behaviors. To obtain a sample of COVID-19-bereaved individuals, we opted for online data collection, given the large number of people accessible online. Due to collecting data online, we were not able to observe behaviors directly. For example, we relied on self-reported time spent in nature rather than direct observations of the time an individual spent in nature. However, we employed established measures (e.g., GAD-7) in an attempt to obtain precise data via self-report. Future research may address this limitation by employing different methods of study variable data collection (e.g., ecological momentary assessment for time spent in nature, clinician evaluation of complicated grief presence), perhaps with bereaved individuals locally available to researchers.” 

Comment 6: “The NDVI is very interesting but the resolution at 250m may not have identified more subtle variations in vegetation.”

Response 6: We have added this limitation of the study in 4.2. Limitations & Future Direction. We state: “Another potential limitation of the current study is the resolution at which surrounding greenness images were taken. The resolution at which surrounding greenness images were taken may have limited the precision by which we were able to estimate participant surrounding greenness. Namely, a higher resolution than the 250m resolution of surrounding greenness images used in the current study could identify more subtle variations in vegetation. Therefore, researchers may consider using or constructing methods for surrounding greenness estimation with improved resolution.”

Comment 7: “Describe in more detail the missing data of the participants.”

Response 7: We have added more detail in describing the missing data of participants as seen in section 3. Results. We now state: “Not all participants completed all measures in full. In some cases, participants did not provide any response on measures of anxiety (n=1), depression (n=1), time spent in nature (n=2), or U.S. postal code (n=3). We were not able to locate one participant’s postal code, therefore, constituting an additional missing value for surrounding greenness. In some cases, participants answered all but one question on measures of complicated grief (n=1), general grief (n=4), anxiety (n=2), depression (n=2), feeling connected to nature (n=3), and time spent in nature (n=1). Further, two participants answered all but two questions on the measure of general grief. Nevertheless, we retained these participants when they had provided sufficient data (i.e., answered most questions for a measure) for a particular set of analyses. We did not substitute or impute data in cases of missing data.”

Comment 8: “Other questions to be resolved are how the possible variability of interpreting the questions was managed.”

Response 8: With any type of self-report data collection, it may be difficult to ensure low variability in question interpretation. However, we included established measures from the larger psychological literature to capture our key constructs. The Brief Grief Questionnaire (BGQ) is a 5-item, validated self-report tool that is frequently used to assess complicated grief in cross-sectional studies (e.g., Yamaguchi et al., 2017; Aoyama et al., 2017; Miyajima et al., 2014). Similarly, the Bereavement Experience Questionnaire (BEQ) is a 24-item questionnaire that is frequently used to assess grief, demonstrating adequate internal consistency and construct validity (e.g., Guarnaccia & Hayslip, 1998; Hayslip Jr., Pruett, & Caballero, 2015; Murrell et al., 2017; Francis et al., 2015). The Generalized Anxiety Disorder Screener (GAD-7) is a highly used measure for assessing anxiety symptoms and severity (e.g., Mattila et al., 2020; Haffart, Johnson, & Ebrahimi, 2021; Hazumi et al., 2022; Palgi et al., 2020). Further, the PROMIS Short Form v1.0 Depression 8a has commonly been used to assess depressive symptoms (e.g., White Makinde et al., 2024; Schalet et al., 2020; Graboyes et al., 2023). A common measure for assessing the degree of emotional connection to nature is the Connectedness to Nature Scale (CNS; Mayer & Frantz, 2004; e.g., Mayer et al., 2009; Markowitz et al., 2018; Bratman et al., 2015). For time spent in nature, we included items modified from Beyer and colleagues (2018) and Larson and colleagues (2019). We used similar wording and scale anchors to Beyer et al. (2018) and Larson et al. (2019). Finally, we estimated greenness surrounding participants’ residential areas based on a single-item, self-reported U.S. postal code. 

Round 2

Reviewer 1 Report

Comments and Suggestions for Authors

The authors responded to the raised comments